# Targeting Neoantigens in Pancreatic Ductal Adenocarcinoma

**DOI:** 10.3390/cancers16112101

**Published:** 2024-05-31

**Authors:** Gurkaranjot Singh, Drew Kutcher, Rajeshwar Lally, Vikrant Rai

**Affiliations:** Department of Translational Research, College of Osteopathic Medicine of the Pacific, Western University of Health Sciences, Pomona, CA 91766, USA; gurkaranjot.singh@westernu.edu (G.S.); drew.kutcher@westernu.edu (D.K.); rajeshwar.lally@westernu.edu (R.L.)

**Keywords:** pancreatic ductal adenocarcinoma, chemotherapy, immunotherapy, treatment resistance, neoantigens, targeted therapy, personalized therapy

## Abstract

**Simple Summary:**

Pancreatic ductal adenocarcinoma (PDAC) is one of the leading causes of cancer deaths in the United States due to its late-stage diagnosis. Currently, there are no early diagnostic tests. The best treatment for PDAC is surgery, but by then, the cancer has already spread, making it difficult to perform. This study aims to address the potential of targeting neoantigens, unique cancer-specific proteins that trigger an immune response, as potential immunotherapy targets. However, PDAC develops characteristics such as immune cell avoidance, creating an environment suppressing immune cells, and converting cells to become unstable. Targeting neoantigens could revolutionize PDAC therapy, offering better survival rates. Though trials are still in the early stages, they do offer a greater understanding and potential for fighting cancer. This article hopes to shed light and pave the way for innovative treatments to improve the prognosis of PDAC patients.

**Abstract:**

Pancreatic ductal adenocarcinoma (PDAC) is the most common type of pancreatic cancer and is currently the third leading cause of cancer-related death in the United States after lung and colon cancer. PDAC is estimated to be the second leading cause of cancer-related death by 2030. The diagnosis at a late stage is the underlying cause for higher mortality and poor prognosis after surgery. Treatment resistance to chemotherapy and immunotherapy results in recurrence after surgery and poor prognosis. Neoantigen burden and CD8+ T-cell infiltration are associated with clinical outcomes in PDAC and paucity of neoantigen-reactive tumor-infiltrating lymphocytes may be the underlying cause for treatment resistance for immunotherapy. This suggests a need to identify additional neoantigens and therapies targeting these neoantigens to improve clinical outcomes in PDAC. In this review, we focus on describing the pathophysiology, current treatment strategies, and treatment resistance in PDAC followed by the need to target neoantigens in PDAC.

## 1. Introduction

An estimated 611,720 people will die of cancer in the US in 2024, with pancreatic cancer accounting for 51,750 [1]. It is currently the third leading cause of cancer-related death in the United States after lung and colon cancer [2]. It is projected to become the second leading cause of cancer-related death by 2030, overtaking colon cancer [3]. Pancreatic duct adenocarcinoma (PDAC) is the most common type of pancreatic cancer, accounting for approximately 85% of pancreatic cancer cases. PDAC originates in the cells lining the ducts of the pancreas, which are responsible for producing digestive enzymes that aid in the digestion of food. PDAC has a high likelihood of metastasis and mainly affects the liver, peritoneum, and lungs [4]. PDAC is notoriously aggressive and often diagnosed at an advanced stage, contributing to its poor prognosis. The 5-year survival rate of PDAC is less than 10%, making it one of the deadliest cancers [5]. Only 24% of people survive 1 year, and 9% live for 5 years [6]. PDAC is more common in men than women, with the median age of diagnosis being 70 years for both [7]. The exact cause of PDAC is unknown, but many modifiable risk factors are associated with the development of this cancer. Modifiable risk factors include smoking, alcohol, dietary factors, pancreatitis, obesity, viral hepatitis, occupational exposure, physical activity, and altered microbiome [8].

Genomic sequencing studies on PDAC found four main driver mutations of the cancer (*KRAS*, *CDKN2A*, *TP53*, and *SMAD4*) [9]. KRAS mutation has been found in >90% of PDAC cases [10]. *KRAS* is a *RAS* gene that produces proteins to regulate cell division. In its standard physiological configuration, the KRAS protein adopts an inactive conformation when bound to guanosine diphosphate (GDP), thereby exerting an inhibitory effect on cellular proliferation. However, a mutation, typically a missense mutation, will cause it to be GTP-bound and active. This causes the cell to be overstimulated by signaling pathways to drive cancer growth [11]. *TP53* is the second most common mutation in PDAC, present in around 70% of all PDAC cases. *TP53* encodes the p53 protein, a tumor-suppressive transcription factor [12]. A mutation, typically a missense mutation, leads to uncontrolled cell division [13]. Gain-of-function (GOF) of *TP53* allows cancer cells to reprogram immune cells towards a pro-tumorigenic phenotype. At the same time, loss-of-function (LOF) mutations trigger systemic inflammation and weaken the T-cell anti-tumor response [12]. TP53 GOF was associated with a substantially worse prognosis than TP53 non-GOF in PDAC patients [14].

There is a lack of simple, early detection methods, and PDAC is typically diagnosed at a late stage because symptoms do not appear until the disease has progressed and metastasized to distinct sites—around 50% of diagnosed patients present with metastatic disease. Surgical resection with chemotherapy provides the best treatment option for PDAC and is beneficial in patients whose cancer cells have not spread to critical abdominal vessels and adjacent organs [15]. At diagnosis, PDAC is usually at least 2–4 cm in diameter and usually has already infiltrated the surrounding structures and lymph nodes. Unfortunately, due to the late-stage diagnosis, just 10–15% of patients are candidates for surgery, which is associated with a higher 5-year survival rate of around 25% [16]. Abdominal pain is the most frequently reported clinical symptom. However, patients are commonly asymptomatic in the early stages [4]. Other symptoms include weight loss, pruritus, and jaundice, which are typically present when the tumor metastasizes [17]. Not much advancement has been made with PDAC management. Aside from the rare instance of surgery, polychemotherapy is the most common form of treatment [18]. Polychemotherapy for PDAC typically uses FOLFIRINOX and nab-paclitaxel/gemcitabine [19]. However, unlike most cancers, PDAC is chemotherapy-resistant. Even with the most effective polychemotherapy protocols, the median overall survival of stage IV patients is only 11.1 months [18,20]. With the difficulty of detecting PDAC and the ineffectiveness of current treatment, research efforts have focused on areas such as immunotherapy, especially as PDAC incidence rates continue to rise. For this, it becomes important to have novel targetable factors, and the concept of targeting neoantigens in PDAC is an emerging area of research.

## 2. Targeting Antigens: Immunotherapy in PDAC

Cancer immunotherapy has emerged as a promising approach to treatment for multiple malignancies. It offers the potential for increased response and improved survival rates. Cancer cells produce cytokines and chemokines that attract immune cells [21]. The basis of immunotherapy is the mechanism by which T cells (CD4+ and CD8+) recognize the tumor antigens presented by major histocompatibility complexes (MHC) with the help of antigen-presenting cells (APCs). Once identified, T cells kill the cancer cells [22,23]. T cells are the primary component of adaptive immune response. The two types of T cells are CD4+ helper T cells and CD8+ cytotoxic T cells. CD4+ T-cell receptors bind to MHC Class II molecules, leading to activation of B cells and cytotoxic T cells [24]. CD8+ T-cell receptors bind to MHC Class I molecules, leading to the death of that cell. However, tumors can evade immune response (immune evasion) by various methods, including rapid and spontaneous proliferation, T-cell exhaustion, and inhibiting the immune system [25]. Chimeric antigen receptor (CAR) T-cell therapy has had significant success in the treatment of hematological malignancies but has limited efficacy in pancreatic cancer [26]. CD318, TSPAN8, and CD66c are emerging targets for CAR T-cell therapy [27].

The improved understanding of antigens has ushered in a new era of potential treatments. Cancer vaccines involve the administration of specific tumor-associated antigens (TAAs) combined with activated dendritic cells. This vaccine stimulates the patient’s adaptive immune system, leading to improved regression of the tumor and its hopeful eradication [28]. Cancer vaccines are unfortunately ineffective against malignant cancers due to the constant changes in the cancer microenvironment and the presence of cancer heterogeneity. In those situations, constant vaccinations are required, but the patient may not have the appropriate time to establish a well-developed immune response [29]. There are only two approved preventive cancer vaccines (HPV and Hepatitis B) and three approved therapeutic cancer vaccines (prostate cancer, bladder cancer, and melanoma) [30]. These vaccines show promise, as in the case of the prostate cancer vaccine, which showed a greater than two-fold increase in prostate cancer-specific T cells in 57% of patients [31]. There was also a reduced risk of death in 41% of patients taking the vaccine [32]. Though vaccines do not completely kill the tumor, it does allow patients to live longer. The studies for the use of oncolytic viral therapy and vaccine therapies including KRAS vaccines, Gastrin vaccines, Telomerase vaccines, Heat-shock protein (HSP) peptide complex-based vaccines, Survivin-targeting vaccines, Listeria-based vaccines, MUC-1 targeting vaccines, and Dendritic cell-based vaccines suggest the therapeutic efficacy of some of these vaccines while more studies are needed to verify the efficacy of these vaccines [33]. Further, immunization of PDAC patients with multi-antigen targeted DNA vaccine may be useful in eliminating the chemoresistance in PDAC [34].

Immune checkpoint inhibitors (ICIs) have also emerged as a potential treatment for PDAC. ICIs take advantage of the patient’s immune system to fight the tumor. Programmed cell death 1 (PD-1) and cytotoxic T lymphocyte antigen 4 (CTLA-4) are co-inhibitory receptors expressed on T cells that negatively regulate T-cell-mediated immune responses. Tumor cells exploit PD-1 and CTLA-4 to cause tumor tolerance and T-cell exhaustion. However, with ICIs, they attach to these co-inhibitory receptors to reactivate their response to tumor cells [35]. Monoclonal antibodies, a highly specific biologic that targets extracellular proteins and intracellular oncoproteins, have shown their efficacy in various cancers but is limited in PDAC. Targeting EGFR, mesothelin, mucins, carcinoembryonic antigens, and intracellular proteins using monoclonal antibodies is an emerging field under research [36].

This suggests the need for a multifaceted, personalized approach involving cellular immunotherapy treatment to improve pancreatic cancer outcomes. Further, targeting the suppressing cells including regulatory T cells (Tregs), regulatory B cells (Bregs), and antigen-presenting cells most potently suppressing immune response in PDAC may improve the clinical outcome [37]. Treatment resistance in PDAC that can be innate or acquired is the primary contributing factor to treatment ineffectiveness and poor survival.

## 3. Treatment Resistance in PDAC

The poor prognosis of PDAC, aside from the late-stage diagnosis, is due to the ineffectiveness of treatments, including chemotherapy and radiotherapy. Chemoresistance in PDAC is due to T-cell exhaustion, a diminished state of T cells in a chronic environment leading to increased expression of inhibitory receptors, decreased effector cytokines, and compromised cytotoxicity activity on tumor cells. PD-1 is highly expressed in this case, leading to the loss in the ability to fight cancer [38]. Another factor contributing to chemoresistance in PDAC is the desmoplastic stroma of the tumor microenvironment (TME) consisting of immunosuppressive cells, pancreatic stellate cells (PSCs), endothelial cells, fibroblasts, and immune regulatory factors [39]. This stroma creates a barrier for the tumor, making it difficult for T cells to attack [39]. PSCs comprise a majority of the stroma, around 50% [40]. PSCs have a variety of roles, including ECM production, cell proliferation, invasion, and drug resistance (Figure 1). PDAC has been shown to have an increased amount of Tregs in its TME, which helps drive its immune evasion. Treg is an immunosuppressive cell which would allow the tumor cell to continue growing in this scenario [41] (Figure 1).

Many studies have found a relationship between the chemoresistance activity of PDAC and the epithelial–mesenchymal transition (EMT). EMT is pivotal in driving tumor invasion, metastasis, and resistance to conventional chemotherapies. There are three types of EMT, Type 1, 2, and 3, with Type 3 being part of metastatic cancer [42]. Epithelial cells are tightly bound and have close contact with the basement membrane via basal–apical polarity, providing a stable structure. This stability is due to cadherin molecules on the epithelial cells called cell surface E-cadherins [43]. E-cadherins create adherens junctions and play a vital role in cell polarity [42]. The loss of E-cadherins is a hallmark of EMT. Several changes occur as it becomes mesenchymal, including loss of E-cadherins and gaining N-cadherins and vimentin [44]. Activation of EMT is accomplished by the activation/overexpression of EMT translational factors such as SNAIL and TWIST [44]. However, studies have shown that ZEB1 seems to be a key player in driving metastasis of PDAC [45]. ZEB1 is repressed by Rb1, which is part of the Ras family. Due to the mutation of KRAS, the Rb1 pathway becomes inactivated. This leads to ZEB1 being expressed, thus inducing EMT [46] (Figure 1). Mesenchymal cells are not polarized, have elongated morphology, and do not form cell junctions; instead, they only have weak adherence to neighboring cells. This allows the cancer cell to migrate and invade [42].

One of the main strategies to combat PDAC is the utilization of gemcitabine, a chemotherapy-agent part of the first-line treatment. Gemcitabine works by inhibiting cancer cell proliferation and promoting apoptosis via activation of the AMPK/mTOR pathway [47]. Gemcitabine is a pro-drug that undergoes enzymatic conversion by nucleoside kinases, resulting in the production of gemcitabine diphosphate (dFdCDP) and triphosphate (dFdCTP). dFdCTP competes with dCTP during the G1/S phase, thereby halting the cell cycle [48,49]. Though it is effective at halting DNA synthesis, gemcitabine efficacy is hampered by the rapid development of resistance in PDAC cells [47]. A gene expression microarray analysis of 165 drug-resistant genes found to be overexpressed in PDAC cells revealed that these genes were diverse and multifaceted in their mechanisms of developing resistance. Some genes alter the TME, making it difficult for drugs to penetrate. Upregulation of ATP-binding cassette (ABC) transporters allows for increased efflux of chemotherapeutic agents. Some miRNA transcripts were found to silence tumor suppressor genes while overexpression of intracellular detoxifying enzymes such as aldehyde dehydrogenases inhibited the effects of gemcitabine [47,48,50].

A change in the tumor microenvironment via cancer-associated fibroblasts (CAFs) in PDAC, with one of the densest cancer stroma, increases drug resistance. This denser and stiffer ECM or stroma is a result of CAF-derived ECM remodeling. The CAF cells secrete proteins to induce desmoplasia and fibrosis of pancreatic tissue, resulting in the formation of a physical barrier to drugs such as gemcitabine. The CAFs also alter the microenvironment by providing nutrients to sustain cancer cell growth despite tumor conditions being hypoxic and undernourished [51,52]. The CAF-induced desmoplasia and hypoxia not only make it more difficult for gemcitabine physically to penetrate, but it also diminishes the effect gemcitabine can have on the PDAC cells. Hypoxia leads to the stabilization of the HIF-1α receptor, which is a regulator of glucose metabolism [48]. A metabolic shift to increased glucose demand in PDAC cells contributes to gemcitabine resistance. The increased glucose addiction results in increased pyrimidine biosynthesis, further resulting in increased levels of deoxycytidine triphosphate (dCTP). Gemcitabine products act as competitive inhibitors in DNA synthesis and higher concentrations of dCTP can overcome the inhibitory effects resulting in increased resistance [48].

ATP-binding cassette (ABC) transporters found in all organisms are active membrane transporters responsible for transporting compounds such as xenobiotics against their concentration gradient and are often a core component underlying many mechanisms of drug resistance [47,53]. Other than gemcitabine, some of the common PDAC therapeutics are oxaliplatin (OXA), 5-fluorouracil (5-FU), irinotecan (IR), and NAB-PTX. These therapeutics, except IR, have associated ABC transporters responsible for their efflux out of the cell. The ABC transporters, breast cancer-resistant protein (BCRP) and multidrug resistance protein 4 (MRP-4), are responsible for the efflux of glutathione-conjugated 5-FU, gemcitabine, and NAB-PTX, while P-glycoprotein is responsible for the efflux of OXA. Along with the primary overexpressed ABC cassettes, ABC cassette synthesis-associated genes such as ABC4/11, ABCC1/3/3/10, and ABCG2 were also overexpressed in PDAC cells [53,54].

Another mechanism of resistance is conferred via the overexpression of aldehyde dehydrogenases, specifically the isoform ALDH1A1 in PDAC cells. Aldehyde dehydrogenases (ALDH) are a family of 17 isoforms found in higher quantities in rapidly dividing and metabolically active cells. Often, active metabolism leads to the production of toxic aldehydes, which ALDH oxidizes to convert them into carboxylic acids [47,55,56]. In recent years, overexpression of ALDH was found to be associated with multiple cancers: pancreatic, breast, colon, lung, liver, and ovarian [55]. Increased enrichment of ALDH1A1 in the MIA PaCa-2 cell line exposed to increasing concentrations of gemcitabine and attaining gemcitabine resistance signifies the role of ALDH1A1 in tumor resistance to gemcitabine. Further, siRNA-based ALDH1A1 knockouts revealed increased rates of apoptosis in knockouts when exposed to gemcitabine [56]. This suggests that ALDHs play an important role in PDAC chemotherapy resistance. The various mechanisms described above suggest that tumor heterogeneity, changing gene expression profile with tumor growth, and tumor microenvironment play a crucial role in chemotherapy resistance and there is a need to identify novel targets for combination therapy to improve clinical outcomes. Neoantigens playing a critical role in tumorigenicity may be attractive therapeutic targets.

## 4. Neoantigens

When normal cells become mutated and undergo the process of tumorigenesis, LOF, and GOF mutations impact tumor suppressor genes and oncogenes, respectively. As a result, the cells undergo rapid replication, all the while accumulating additional passenger mutations, which can lead to the production of new protein variants unknown to the immune system [57,58,59]. Due to recent advances in proteome and genome sequencing, increasingly more markers are being found on cancers, specific to only those cancer cells. These markers come in two categories: tumor-associated antigens and tumor-specific antigens [57,58,59]. A subset of tumor-specific antigens that are membrane-bound and immunogenic are called neoantigens. In the past few years, many cancers including PDAC have been shown to express these unique markers. This is important because immunological therapies can be developed which can help the immune system identify and destroy cancers specifically without harming other cells in the body.

### 4.1. Neoantigen Generation

Neoantigens can be formed either through genomic mutations, dysregulated post-transcriptional modification, or a viral-derived tumor antigen from carcinogenic phages [58,59]. For genomic mutations, only a select few types of mutations result in the neopeptide being processed and presented on the cell surface via MHC class I molecules. The most common mutation is a single nucleotide variant (SNV) followed by insertion/deletion and gene fusions [59]. SNVs can generate neoantigens due to their relatively small change—no shifts in reading frame resulting in transcript degradation before translation. While they are the most frequently seen mutation, their immunogenic capacity is limited due to their similarity to self-antigens. The number of SNV-predicted neoantigens has been shown to have low (around 3%) immunogenicity [60]. In the tumors with high intratumor heterogeneity, the immune escape may be due to the loss of strong-binding neoantigens produced by SNVs [61]. Insertion/deletions (INDELS) can also create neopeptides, but are more frequently responsible for shifts in the reading frame and are self-selected. Though INDELS generate fewer neoantigens, research suggests that INDEL-based neoantigens possess greater MHC-1 binding affinity and immunogenic potential than SNVs [62]. INDELs create distinct peptides that have greatly enhanced recognition by the immune system and thus are highly immunogenic [62]. However, decay or degradation of neoantigens via the nonsense-mediated decay pathway is a concern, but some frameshift INDELS may escape degradation and elicit immunogenicity [63]. Another type of mutation that produces neoantigens is gene fusions, specifically chromosomal inversion or translocation. Gene fusions, though fewer in number, result in the greatest immunogenic neoantigens when compared to SNV and INDELS [57]. Gene fusion-derived neoantigens are more immunogenic because of the presence of multiple targets per mutation and their wide distribution across cancer types [64]. The FusionNeoAntigen resource may help develop fusion-based immunotherapies and can also be used as a prognostic biomarker of immune checkpoint blockade. Fusion-based neoantigens, which are more frequent in tumors with suppressive immune environments, may also be used to develop tumor vaccines, adoptive cell therapies, and modulation of the tumor microenvironment [65]. Additionally, neoantigens may also be produced by mutations due to exon–exon junctions, intron retentions, and alternative splicing events [66].

Transcriptomic variants also generate neoantigens due to errors in alternative splicing and alternative polyadenylation events (APA). There are three types of alternative splicing mutations: cis-acting, trans-acting, and nonsense-mediated degradation (NMD) impairment. Cis-acting mutations during alternative splicing occur due to an inherent defect in the regions of the mRNA that control splicing. Most often, mutations can result in retained introns or missing exons acting akin to INDELs. Trans-acting mutations occur when a mutation impacts proteins involved in the splicing process, such as small nuclear ribonucleoproteins (sNRPS), and can lead to similar retentions of introns as cis-acting [47]. NMD is a regulatory mechanism responsible for the degradation of aberrant RNA transcripts. The UPF1 gene encodes for the core component of NMD, and mutations in this gene have been shown to increase neoantigen production greatly as errant mRNAs, undergo translation to produce neoantigens instead of being degraded [47,67]. Polyadenylation involves adding a poly(A) to the 3’ of mRNA. Usually, this occurs in an untranslated region called UTR, but researchers have found an increased prevalence of abnormal poly(A) additions in cancer cells, resulting in truncated proteins. These truncated proteins are also a potential target for neoantigen-based cancer immunotherapy [47,67]. Viral-derived neoantigens have been shown as promising targets in tumors caused by viruses. Oncogenic viruses insert their DNA into the cellular genome to produce viral proteins, which are inherently immunogenic. Many cancers, such as cervical cancer from HPV, have seen regression in cancer due to high-affinity TCR responses [47,68].

### 4.2. Characteristics of a Good Neoantigen

Having discussed the various ways neoantigens are generated in cancer cells, the next step would be to characterize and compare the neoantigens to determine which is best for immunotherapy. This characterization step is crucial, as neoantigens are not only specific to a type of cancer but can also be specific to a specific subset of the population diagnosed with that cancer [47,69]. Thus, neoantigen screening would become paramount to developing personalized treatments for individuals. There are several factors to consider when selecting a neoantigen; they need unique peptides generated via the mechanisms discussed above, the neoantigen and MHC interaction needs to be strong, and the peptide-bound MHC (pMHC) and TCR affinity should be strong as well [47,69,70]. The neoantigen mutations are characterized by whole exome sequencing (WES) in combination with RNA-seq. Once the mutations in the specific cancer cell of choice have been mapped, high-throughput immunogenic techniques can be used to confirm whether mutant peptides are bound to the patient’s own MHC I subtype [70]. Mass spectrometry confirms the structures of the peptides isolated via MHC immunoprecipitation. The selected neopeptides that have been MHC restricted are then put through MHC multimer assays to determine the degree of CD8 T-cell activation [71].

### 4.3. What Are the Current Limitations of Neoantigens?

Though immunotherapy has many advantages such as being highly selective with tumor cells, it still faces some of the same challenges that chemotherapy also faces. These challenges as described earlier include TME penetration issues, therapeutic pressure, tumor heterogeneity, and immune evasion [47]. The tumor microenvironment for solid cancers is remodeled by CAFs. CAFs are responsible for remodeling the ECM to promote fibrosis and scarring. These lesions functionally compartmentalize the solid cancers until metastasis, protecting them from both the immune system and chemotherapeutics alike [47,72]. When neoantigen therapy is selected with a desired epitope, the immune system hopefully generates cytotoxic CD8 T cells which recognize cancerous cells. However, these CD8+ T cells would have difficulty exerting their full effects if they physically cannot interact with the ligands on tumors due to the TME [72].

Another current limitation of neoantigens revolves around the concept of tumor heterogeneity, which is both spatial and temporal in its characteristics. Spatially, the solid tumor becomes separated into two categories, immune-hot and immune-cold. The hot tumor portions are immunogenic and would be influenced, and as a result, the cold tumor cells would undergo positive selection. Even if all cells in the tumor are ”hot,“ due to the prolonged exposure time of the immunotherapy and the inherent genetic instability of cancer cells, the neoantigens selected for the immunotherapy could become obsolete for a newly mutated population of cancerous cells [73]. Cancer cells themselves also have inherent immune evasion properties via the overexpression of programmed cell death protein 1 ligand (PD-L1). PD-L1 binds to PD-1 on T cells to inhibit TCR signaling and prevent cytotoxic effects [47,74]. PD-L1 normally expressed on the surface of T lymphocytes, B lymphocytes, DCs, and macrophages is also highly expressed on the tumor cell surface, and this causes T-cell exhaustion, immune tolerance, and immune evasion [75]. Further, upregulation of PD-L1 on antigen-presenting cells due to increased cytokine expression like IFN-γ inhibits T-cell activation and contributes to immune evasion [76]. A loss, alteration, or downregulation of MHC-1 expression on the cancer cell surface may also cause immune evasion [77]. In the case of non-small-cell lung cancer cells, upregulated expression of transcriptional coactivator MRTF-A by TGF-β further activating NF-κB/p65 axis-mediated PDL1 transcription and expression also contributes to immune evasion [78]. Some current therapeutics are investigating antibodies that can bind to either the PD-1 or PD-1L to prevent their association, thereby allowing the cytotoxic effects to take place [74].

Another immune checkpoint includes CTLA-4. CTLA-4, expressed on T cells, plays a role in downregulating immune response by binding to CD80 on antigen-presenting cells to reduce the availability of the ligand against CD28, which upregulates the immune response. PDAC engages with CTLA-4 to inhibit T-cell activity and continue tumor growth [79]. In a murine model of PDAC, inhibition of CTLA-4 was not found to be sufficient to increase CD8+ T-cell infiltration in the tumor microenvironment suggesting its limitations [79]. Targeting CTLA-4 limits the priming of naïve T cells; however, it also attenuates direct anti-tumor T-cell activity in the effector phase, in parts, by decreasing suppressive Tregs [80]. These limitations may be the reason for lacking a demonstrable efficacy of targeting CTLA-4 in PDAC.

### 4.4. Neoantigens in PDAC

A significant characteristic of cancer cells is their unstable genome due to its high mutation rate. As a result of these somatic mutations, cancer cells produce proteins that are not expressed by normal cells. These tumor-specific proteins are called neoantigens [81]. Neoantigens are the result of mutations occurring in tumor DNA. They may also play a crucial role in developing treatments to assist the body in mounting an immune response against the cancer cells presenting these antigens. Due to PDAC having a lower mutation rate as compared to other cancers, they produce fewer neoantigens [82]; however, a median of 38 neoantigens per PDAC tumor has been reported [83]. An average of 35 neoantigens per PDAC tumor is markedly less than more immunogenic cancers, like non-small-cell lung cancer and melanoma (112 and 370 neoantigens on average, respectively) [84].

These neoantigens are specific to individuals, making it a promising avenue for therapeutic development. Determining the neoantigens of a patient requires knowing the individual’s HLA type, tumor mRNA expression, germline, and tumor DNA. Current technology uses whole genome microarrays and RNA-seq to gather all this information [85]. Specific neoantigens can vary among patients; however, studies have found common mutations as well as neoantigens associated with PDAC.

RNA-seq is a tool that involves isolating all the RNA from a sample at a certain time point. The isolated RNA is then converted to complementary DNA (cDNA) via reverse transcription. The cDNA fragments are also modified by the addition of adapters, which allow the fragments to bind to complementary oligonucleotides on a flow cell. Next-generation sequencing, such as Illumina sequencing, uses fluorescent nucleotides to sequence the RNA with high fidelity, thus providing insights into the type and expression levels of the tumor. Whole genome microarrays work similarly, but instead of sequencing, the cDNA binds to already known sequences of genes, such as HLA serotypes. This process is much faster than sequencing and provides information on what genes are present in the individual. Mapping the tumor-specific genetic abnormalities of the tumor and normal DNA while comparing them helps in identifying the neoantigens [85,86,87]. In the context of identifying neoantigens in PDAC, variants detected in tumor microarrays undergo analysis using epitope prediction algorithms. These algorithms aim to identify peptide sequences capable of binding to the patient’s HLA alleles [85].

Mutations in the *KRAS* oncogene are a significant driver of pancreatic cancers, as it is associated with 85–90% of pancreatic tumors (of which approximately 85% of all pancreatic cancers are PDACs) [88,89]. The *KRAS* gene is crucial in initiating and maintaining pancreatic tumors and is responsible for encoding a member of the Ras family of small GTPases [88]. The Ras signaling pathway is suspected to be a key oncogenic driver of PDAC [88] The most frequent mutations seen in the *KRAS* gene include G12VD (31%), G12V (31%), and G12R (21%). Additionally, 4% of patients with PDAC had multiple concurrent mutations of the *KRAS* gene, which could be further seen in different cells of the same tumor [88]. Mismatch repair–deficient cancers (dMMR) generate a large number of neoantigens; however, dMMR tumors represent only 1% of PDAC cases [90]. With the nature of PDAC being a less immunogenic cancer, fewer higher quality antigens in addition to strong T-cell activity have been associated with PDAC patients with longer survival times [84]. Neoantigens can be separated into shared and personalized antigens. Shared neoantigens are mutations that are seen in different cancer patients. A study examining shared neoantigens in pancreatic cancer patients found 10 neoantigens that were shared in around 50% of cases. KRAS G12D was the most frequent mutation and was identified in 32.42% of pancreatic cancers assessed in the study [91]. BRAF V600E HLA*A03:01 restricted peptide KIGDFGLATEK was the most frequent neoantigen in PDAC, which covered 1.51% of tumor patients (117 of 7748). They also noted the ten most frequent shared neoantigens, which covered approximately 50% of pancreatic cancer patients [91]. Personalized neoantigens are unique mutations that are specific to the individual and are targets of personalized therapy. Bailey P et al. [92] concluded from a systematic study identifying target neoantigens for PDAC immunotherapy that the *KRAS* codon 12 mutations were the most common genetic mutation. Due to these mutations, the corresponding peptide was predicted to produce immunogenic neoantigens. This study found that nearly all PDAC cases express candidate neoantigens. Additional mutations that were found to generate potential neoantigens for multiple patients included KRAS Q61H, TP53 R273N, TP53 R282W, MT-ND4 A318T, FCGBP A2493V, RBM12 P693S, and RET A1019V. A complete neoantigen landscape of PDAC has been summarized [92].

Another study reported that the *MUC16* gene in PDAC patients contained neoantigens in more than 15% of all patients. It was also found to have a four-fold higher frequency of neoantigens in long-term survivors of PDAC compared to short-term survivors. It should be considered as a candidate immunogenic hotspot in PDAC. In addition to the high frequency of *MUC16* neoantigens, increased frequency in *KRAS* and *TP53* genes was also detected. A major conclusion was that the quality of neoantigens is an essential biomarker for immunogenic tumors [83]. An additional study analyzing the quality of neoantigens reported that long-term survivors of pancreatic cancer compared to short-term survivors of pancreatic cancer have up to a 12-fold greater number of activated CD8+ T cells, which are proposed to target immunogenic neoantigens. With this more substantial immune pressure, long-term survivors of PDAC form tumors with fewer neoantigens compared to short-term survivors. A conclusion drawn was the presence of immunoediting in patients with PDAC, which has a low frequency of mutation for cancer and is considered resistant to endogenous immunity. Furthermore, immunoediting, the idea that lymphocytes kill more immunogenic cancer cells to cause fewer immunogenic clones, is present in long-term survivors of PDAC and could be why fewer neoantigens are seen in long-term survivors [83].

### 4.5. Targeting Neoantigens in PDAC

Neoantigens provide a promising approach to fighting cancer. However, PDAC has a lower tumor mutational burden (TMB), around 60 encoded neoantigens. Most cancers have around 100–1500 mutations per MB. Current research is looking to find more neoantigens associated with PDAC via newer assays [93]. Immunotherapy specifically targeting neoantigens is still novel. Vaccines can be used to target neoantigens, however, due to the individual specificity, they require time to develop. Though common neoantigens seen in certain cancers can be used, accuracy is still a potential issue [94].

Currently, unresectable PDAC is most commonly treated with chemotherapies such as gemcitabine and 5-FU [95]. Tumors in PDAC often have low blood vessel density while being densely fibrotic. This restricts drug delivery to tumor cells. Pancreatic cancers can also harbor immunosuppressive TMEs, which decrease their response to immune checkpoint blockade using PD 1 and PD-L1 antibodies [10]. Neoantigen-targeted therapies may be used to stimulate immune responses to recognize and attack cells expressing neoantigens. The current framework for neoantigen-targeted therapies is focused on personalized medicine due to research revealing that tumor mutations are often unique to individual patients; therefore, neoantigens may be strong personalized targets [91]. A systematic approach to developing personalized neoantigen immunotherapies for PDAC can be summarized through four main steps. First is the identification of somatic mutations in PDAC tumors. Second is using bioinformatic algorithms to predict and prioritize neoantigens that will have the highest potential to bind to MHC molecules. Third is the validation of immunogenicity and tumor reactivity through ensuring that selected antigens are capable of eliciting an immune response that specifically targets tumor cells. Fourth is the development of personalized immunotherapies based on the validated neoantigens [91].

Personalized medicine utilizing neoantigens focuses on T-cell-based immunotherapy. These treatments are considered most feasible for patients with a KRAS G12 mutation [96]. Rojas et al. shared findings from phase I trials of cevumeran in combination with atezolizumab (anti-PD-L1), a personalized neoantigen vaccine based on uridine mRNA-lipoplex nanoparticles. They found that cevumeran was tolerable and induced de novo high-magnitude neoantigen-specific T cells in 50% of patients (8 of 16). Ultimately, this study found that PDAC patients treated with adjuvant atezolizumab, autogene cevumeran, and mFOLFIRINOX had substantial T-cell activity that may correlate with delayed PDAC recurrence [82]. Meng et al. found that tumor-infiltrating B cell-derived IgGs could recognize the majority of KRAS G12 mutations found in PDAC. Furthermore, the study proposed that the use of tumor-infiltrating B cells could be a viable molecular blueprint for anti-tumor targeting neoantigens that could be used for personalized monoclonal antibody therapy [97].

Freed-Pastor et al. identified that the CD155/TGIT axis is a key driver of immune evasion in PDAC. Furthermore, they found that predicted neoantigens revealed a subset of PDAC patients with high-affinity neoantigens. This led to the functional integration of a combination therapy consisting of TIGIT/PD-1 co-blockade and CD40 agonism, which elicited anti-tumor response in preclinical models. With these findings, they also called for future studies to evaluate the requirements for high-affinity neoantigens in mediating this response [98]. Zhang et al. summarized various personalized neoantigen-based vaccines and neoantigen-based dendritic cell vaccines for pancreatic cancer as of 2023 [99]. One example is the iNeo-Vac-P01, a neoantigen-based vaccine that enhanced the clinical efficacy of pancreatic cancer through the increase of antigen-specific TCR clones elicited by the vaccine [100]. In phase I trials, the iNeo-Vac-P01, seven patients with advanced pancreatic cancer were enrolled. Neoantigen peptides were identified in each patient by the researchers that were then manufactured and administered to the patients with low tumor mutation burden. Patients were given multiple doses of the personalized neoantigen vaccine concerning the progression of their disease. The vaccine administration was not associated with any severe adverse effects. One patient who received treatment and a 21-month overall survival had a drastic increase from 0% to 100% increase in antigen-specific TCR clones [100].

Personalized neoantigen vaccine NEO-PV-01, alongside chemotherapy and anti-PD-1 therapy in non-squamous non-small-cell lung cancer showed efficacy in increasing immune cell response to the cancer and increased CD4 T cells in the patients provides a promising result with combination therapy [101]. A recent study generated multiple genetically distinct cell lines of KP2, a cell line derived from the KPC model of PDAC treated with oxaliplatin and olaparib (KP2-OXPARPi) and exome sequencing and in silico neoantigen prediction confirmed that some clones/neoantigens are immunogenic and their growth can be attenuated by neoantigen synthetic long-peptide vaccines, while other clones/neoantigens are not immunogenic [102]. These results suggest that tumor heterogeneity may pose a challenge to the success in targeting neoantigens even with combination therapy. Neoantigen-based therapies have shown promising results in the initial trials; however, studies in PDAC are limited and more research on developing personalized neoantigens must be completed before combination therapies can be considered in PDAC. Identification of accurate mutations and high-quality immunogenic neoantigens by integrating DNA sequencing data with RNA sequencing data may be advantageous in improving the therapy and clinical outcomes [87].

Completed and ongoing clinical trials of neoantigen-based therapies for PDAC suggest the importance of targeting neoantigens in PDAC (Table 1).

Dendritic cells (DC) can present neoantigens to T cells and induce a specific immune response. Conventional DCs (cDCs) can become dysregulated and are thus highly correlated with abnormal immune surveillance and hindered response of the immune system to PDAC neoantigens [99]. The quantity and functionality of cDCs in PDAC could also be considered as a biomarker for adaptive immune responses to PDAC tumor neoantigens [103]. Zhang et al. thus concluded that DCs could play a vital role in neoantigens presentation and inducing neoantigen-specific T-cell receptors due to the neoantigen-loaded DC vaccines being able to directly present neoantigens to T cells. An example of a DC vaccine in pancreatic cancer was seen in a phase I pilot study in 2015. This study evaluated the feasibility and efficacy of a DC vaccine pulsed with Wilm tumor gene-1 (WT1) peptide combined with gemcitabine in treating patients with advanced PDAC. They found that the WT1 peptide-pulsed DCGEM could trigger anti-tumor TCRs, but was less effective in treating PDAC with live metastasis and increased inflammatory markers [104].

## 5. Future Directions

Increasing the therapeutic efficacy of neoantigen-based immunotherapies requires a multifactorial approach. Continuing neoantigen identification and prediction models are crucial to identifying neoantigens that can be targeted. Furthermore, the continued study of PDAC neoantigens can reveal potential shared neoantigens between PDAC patients. Doing so could lead to the development of immunotherapies that could be administered faster than personalized vaccines. Increasing therapeutic efficacy can also come from the continued development and trials of multi-epitope vaccines. The optimization of the vaccination form is also needed to enhance therapeutic efficacy. Currently, many treatments of PDAC using neoantigen-based immunotherapies revolve around neoantigen-based vaccines, yet neoantigen-based DC vaccines should also be evaluated as immunotherapy candidates. Furthermore, the importance of evaluating combination therapy to increase the therapeutic efficacy of PDAC is vital, as results previously mentioned have demonstrated optimistic results using neoantigen-based immunotherapies alongside traditional PDAC therapies [99]. The continuation of personalized approaches is also crucial to maximize the therapeutic efficacy of treating PDAC through the development of robust immune responses. Finally, biomarker development and monitoring methods are key to providing insight into the immune responses to treatments and help predict the outcomes of such treatments.

As one of the most lethal cancers, early diagnostic tools and improved treatment options for PDAC are needed. Developing a diagnostic tool to identify early gene mutations seen in PDAC such as the KRAS would prove to be beneficial. Currently, there is only one FDA-approved serum biomarker for PDAC–CA19-9 [105]. A hallmark of PDAC is its heterogeneity. The heterogeneity is what makes PDAC resistant to current treatment modalities, but also makes it unique among patients. Because of its low TMB, it makes it difficult to develop an effective neoantigen for PDAC. Developing therapies focused on restoring and increasing T-cell response would prove to be effective against PDAC’s TME.

## 6. Conclusions

PDAC has a grave prognosis due to its delayed diagnosis, and chemoresistance and recurrence add to this predicament. Based on the existing literature, targeting neoantigens seems promising, and novel therapeutic strategies are needed to improve the clinical outcome. Further, combination therapies including chemotherapy, radiotherapy, and immunotherapy-targeting neoantigens may improve clinical outcomes, though warrant more research.

## Figures and Tables

**Figure 1 cancers-16-02101-f001:**
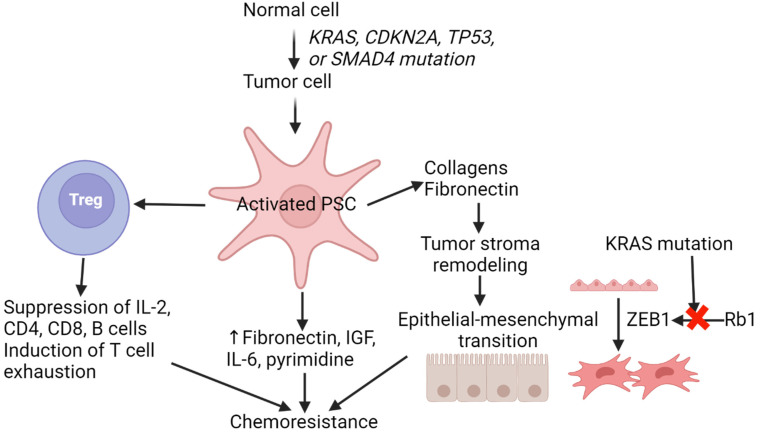
Chemoresistance in pancreatic ductal adenocarcinoma. Increased pro-inflammatory cytokines, extracellular matrix remodeling of the stroma, epithelial–mesenchymal transition, and mutations contribute to chemoresistance in pancreatic ductal adenocarcinoma. Interleukin (IL), cluster of differentiation (CD), insulin-like growth factor (IGF), and pancreatic stellate cells (PSCs).

**Table 1 cancers-16-02101-t001:** Clinical trials of neoantigen-based therapies on PDAC.

Intervention/Treatment	Phase	Number of Subjects	Status	Clinical Trial ID
KRAS peptide vaccine	Phase 1	25	Recruiting	NCT05013216 (PDAC)
KRAS peptide vaccine, Nivolumab, Ipilimumab	Phase 1	30	Recruiting	NCT04117087 (PDAC)
Neoantigen vaccine with poly-ICLC adjuvant, Retifanlimab	Phase 1	0	Withdrawn	NCT04799431 (PDAC)
Personalized neoantigen vaccine	Phase 1	30	Recruiting	NCT03558945 (PC)
Optimized neoantigen synthetic long-peptide vaccine, poly-ICLC	Phase 1	25	Active, Not Recruiting	NCT05111353 (PC)
Personalized neoantigen DNA vaccine	Phase 1	15	Terminated	NCT03122106 (PC)
Neoantigen peptide vaccine, poly ICLC	Phase 1	12	Terminated	NCT03956056 (PC)
Camrelizumab, SJ-Neo006, Gemcitabine + Abraxane	Early Phase 1	12	Recruiting	NCT06326736 (PC)
Personalized neoantigen tumor vaccine	Early Phase 1	54	Recruiting	NCT05916261 (PC)
iNeo-Vac-P01, GM-CSF	Phase 1	7	Completed	NCT03645148 (PC)
Neoantigen vaccine plus anti-PD1 and chemotherapy	Phase 1	43	Recruiting	NCT06344156 (PC)
XH001, Ipilimumab injection, chemotherapy	Not Applicable	12	Not Yet Recruiting	NCT06353646 (PC)
iNeo-Vac-P01, GM-CSF	Phase 1	20	Recruiting	NCT04810910 (PC)
Adebrelimab, mRNA tumor vaccines	Early Phase 1	30	Not Yet Recruiting	NCT06156267 (PC)
Individualized mRNA neoantigen vaccine (mRNA-0523-L001)	Not Applicable	21	Recruiting	NCT06141369 (PC)
Atezolizumab, RO7198457, mFOLFIRINOX	Phase 1	29	Active, Not Recruiting	NCT04161755 (PC)
Drug: TCR–T-cells injection (GB3010 cells injection)	Early Phase 1	18	Recruiting	NCT06054984 (PC)
Next-generation sequencing (NGS), HLA typing	Observational study	93	Completed	NCT03794128 (PC)
Personalized mRNA tumor vaccine	Not Applicable	24	Unknown Status	NCT03468244 (PC)
IRE + intratumoral Mitazalimab (CD40 antibody) injection	Phase 1	18	Not Yet recruiting	NCT06205849 (PC)
GRT-C903, GRT-R904, Nivolumab, Ipilimumab	Phase 1 Phase 2	39	Completed	NCT03953235 (PC)
Neoantigen specific TCR–T-cell drug product	Observational study	180	Recruiting	NCT05292859 (PC)
Cyclophosphamide, Fludarabine, tumor-infiltrating lymphocytes (TIL), Aldesleukin	Phase 1 Phase 2	20	Recruiting	NCT04426669 (PC)
Neoantigen-specific TCR–T-cell drug product, Aldesleukin (IL-2)	Phase 1 Phase 2	180	Active, Not Recruiting	NCT05194735 (PC)
YE-NEO-001	Phase 1	16	Active, Not Recruiting	NCT03552718 (PC)
Imiquimod, Pembrolizumab, Sotigalimab, synthetic tumor-associated peptide vaccine therapy	Phase 1	150	Recruiting	NCT02600949 (PC)

Clinical Trials of neoantigen-based therapies for PDAC and PC. Trial information was collected from ClinicalTrials.gov. Studies were found with the following search terms used in the ClinicalTrials.gov “focus your search” feature: 1. Condition/disease: “Pancreatic Ductal Adenocarcinoma, PDAC”, Other Terms: “Neoantigen”; 2. Condition/disease: “Pancreatic Cancer”, Other Terms: “Neoantigen”. All studies from results are included. NCT # followed by (PDAC) indicates studies found under the first search result (n = 3). NCT numbers followed by PC were found with the second search result (n = 27). The three PDAC studies from search 1 were included in the 27 articles found in the second search. Three results were observational studies and not linked with phases.

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
