# Peer review of "Targeting Neoantigens in Pancreatic Ductal Adenocarcinoma"

_cancers, 2024, doi:10.3390/cancers16112101_

Round 1

Reviewer 1 Report

Comments and Suggestions for Authors

A good review.  It reviews the existing  targeting antigens,  the mechanisms of treatment resistance, and subsequently looks forward to new antigens in pancreatic cancer. Although this topic is not entirely new, it is rich in content and might contribute to a comprehensive understanding of new treatments for pancreatic cancer.

The table has a lot of content, taking up three pages. Adjustments are needed for ease of reading.

Comments on the Quality of English Language

There are still some minor spelling and grammatical errors that need to be corrected.

Author Response

Comment: A good review.  It reviews the existing  targeting antigens,  the mechanisms of treatment resistance, and subsequently looks forward to new antigens in pancreatic cancer. Although this topic is not entirely new, it is rich in content and might contribute to a comprehensive understanding of new treatments for pancreatic cancer.

Response: Thank you for your comments.

Concern 1: The table has a lot of content, taking up three pages. Adjustments are needed for ease of reading.

Response: Thank you for your suggestion. We have modified the table as per your suggestion.

Concern 2: There are still some minor spelling and grammatical errors that need to be corrected.

Response: Thank you for your comment. We have thoroughly checked the manuscript and have edited the text.

Reviewer 2 Report

Comments and Suggestions for Authors

This review discussed the potential of neoantigen-based immunotherapy for pancreatic ductal adenocarcinoma (PDAC). In this manuscript, the authors discussed the general concept of neoantigen and their role in PDAC's immunosuppressive and heterogeneous tumor microenvironment. This manuscript also includes current clinical trials, the challenges of PDAC's low tumor mutational burden, and potential approaches in vaccine development and combination therapies. Below are some review comments:

Section 4.1 Neoantigen generation

the impact of SNVs, INDELs, and gene fusions on neoantigen immunogenicity could be detailed more thoroughly. i.e. “INDEL-based neoantigens possess greater MHC-1 binding affinity and immunogenic potential than SNVs” in line 262, please explain and provide more detail.

Section 4.3 What are the current limitations of neoantigens

Please explain in detail how PD-L1 expression contributes to immune evasion and its interaction with PD-1. The authors could also discuss other immune checkpoint molecules that are involved and their impact on neoantigen-based therapies.

Section 4.4 Neoantigens in PDAC

The section briefly mentions the use of whole genome microarrays and RNA-seq for identifying neoantigens. Could the authors provide a more detailed description of these techniques and explain how they specifically aid in identifying neoantigens in PDAC?

Section 4.5 Targeting Neoantigen in PDAC

The authors could discuss the primary outcomes measured in studies of neoantigen-based therapies for PDAC and how these outcomes compare to traditional treatment modalities.

Are there ongoing studies investigating the combination of neoantigen-based therapies with other forms of treatment, such as chemotherapy or other immunotherapies? If so, what synergies are anticipated?

Author Response

Comment: This review discussed the potential of neoantigen-based immunotherapy for pancreatic ductal adenocarcinoma (PDAC). In this manuscript, the authors discussed the general concept of neoantigen and their role in PDAC's immunosuppressive and heterogeneous tumor microenvironment. This manuscript also includes current clinical trials, the challenges of PDAC's low tumor mutational burden, and potential approaches in vaccine development and combination therapies. Below are some review comments:

Response: Thank you for your comments.

Concern 1: Section 4.1 Neoantigen generation: the impact of SNVs, INDELs, and gene fusions on neoantigen immunogenicity could be detailed more thoroughly. i.e. “INDEL-based neoantigens possess greater MHC-1 binding affinity and immunogenic potential than SNVs” in line 262, please explain and provide more detail.

Response: Thank you for your comments and suggestions. We have included the text to describe the impact of SNVs, INDELs, and gene fusions on neoantigen immunogenicity detailed more thoroughly (lines 264-286).

Concern 2: Section 4.3 What are the current limitations of neoantigens: Please explain in detail how PD-L1 expression contributes to immune evasion and its interaction with PD-1. The authors could also discuss other immune checkpoint molecules that are involved and their impact on neoantigen-based therapies.

Response: Thank you for your suggestions. We have included the text on how PD-L1 expression contributes to immune evasion in the revised manuscript (lines 346-354). We have also included the text on other neoantigens in PDAC (lines 357-366) in the revised manuscript.

Concern 3: Section 4.4 Neoantigens in PDAC: The section briefly mentions the use of whole genome microarrays and RNA-seq for identifying neoantigens. Could the authors provide a more detailed description of these techniques and explain how they specifically aid in identifying neoantigens in PDAC?

Response: Thank you for your suggestion. We have revised the manuscript and the text related to genome microarrays and RNA-seq for identifying neoantigens in relation to PDAC has been included (lines 384-397 and 512-515)

Concern 4: Section 4.5 Targeting Neoantigen in PDAC: The authors could discuss the primary outcomes measured in studies of neoantigen-based therapies for PDAC and how these outcomes compare to traditional treatment modalities. Are there ongoing studies investigating the combination of neoantigen-based therapies with other forms of treatment, such as chemotherapy or other immunotherapies? If so, what synergies are anticipated?

Response: Thank you for your comments and suggestions. We have revised the manuscript to address the suggestions (lines 500-514). The primary outcomes of targeting neoantigens have been discussed. The literature suggests that phase I clinical trials have promising results, but more research is needed. Similarly, combination therapy has shown some beneficial results and clinical trials are ongoing to investigate the efficacy of combination therapy.

Round 2

Reviewer 2 Report

Comments and Suggestions for Authors

The responses to the review points are well organized, and I have no further questions about this version.